# The Effects of Types of Service Providers on Experience Economy, Brand Attitude, and Brand Loyalty in the Restaurant Industry

**DOI:** 10.3390/ijerph19063430

**Published:** 2022-03-14

**Authors:** Jinsoo Hwang, Jawad Abbas, Kyuhyeon Joo, Seung-Woo Choo, Sunghyup Sean Hyun

**Affiliations:** 1The College of Hospitality and Tourism Management, Sejong University, Seoul 143-747, Korea; jhwang0328@gmail.com (J.H.); khjoo@sju.ac.kr (K.J.); bigseung@empal.com (S.-W.C.); 2Faculty of Management Sciences, University of Central Punjab, Lahore 54000, Pakistan; jawad.abbas@ymail.com; 3School of Tourism, Hanyang University, 17 Haengdang-Dong, Seongdonggu, Seoul 133-791, Korea

**Keywords:** robot server, experience economy, brand attitude, brand loyalty, difference analysis

## Abstract

This study was designed to understand the relationships among the experience economy, brand attitude, and brand loyalty based on the type of service providers, such as robot servers and human servers in the restaurant industry. The data were collected from 296 people who experienced robot servers and from 294 people who experienced human servers and was analyzed through structural equation modeling (SEM), which indicated that the four sub-dimensions of the experience economy: education, entertainment, esthetics, and escapism, positively affect brand attitude, which in turn has a significant positive impact on brand loyalty. In addition, statistical differences were found with the average value of the six constructs based on the type of service providers, such as robot servers and human servers.

## 1. Introduction

As the use of robotics in the hospitality industry continues to increase and evolve, it is important to research the effects and outcomes of implementing robot services. Robots are already fully functioning in the service industry worldwide [1]. The importance of robots is no exception to the restaurant industry. Implementing robots in the restaurant industry offers a variety of benefits for both managers and customers. Robots can make products consistently without error, allowing companies to provide efficient services, reduce waiting times, and increase productivity [2]. Another benefit to using robots in restaurants is that labor, material, and rent costs are continually increasing, while robots offer a way for practitioners to save on labor costs in the long run.

In the service industry, previous studies have consistently argued that when purchasing a product, consumers consider experiential benefits to be more important than practical benefits [3,4]. People want new experiences from a product/service that they have not usually experienced. In turn, they are more likely to have a favorable attitude toward the product/service [5,6]. This theoretical basis suggests that robot services play a significant role in providing customers with experiential benefits [7]. If customers have a memorable experience that they have not usually experienced at a robot restaurant, they are more likely to have a positive attitude toward the restaurant brand. Despite the importance of experiential benefits in robot restaurants, rare attention has been paid to this topic by researchers.

In recent years, robots have been performing many tasks on behalf of humans in the restaurant industry. However, since robots and humans inherently have different characteristics, consumers’ perceptions of restaurants differ depending on whether robots or humans provide services [8]. In particular, after COVID-19, consumers prefer non-face-to-face services to prevent infection [9], which suggested the differences in experiences based on the type of service providers, such as robots and humans. Although there are differences in consumer experiences depending on whether the service provider consists of robots or humans, research related to it is scarce.

In summary, consumer experiences are an important factor in predicting consumer behavior in the service industry [5,6,7], but there are few studies on the background for robotic restaurants. Therefore, this study aims to understand the importance of the experience economy and the effects of its outcome variables in robot restaurants. In addition, since the consumer experiences may vary depending on the service provider [8], this study attempts to find out the difference between robot servers and human servers in the experience economy for the first time in the restaurant industry. Specifically, this study examines the effect of four sub-dimensions of the experience economy, namely education, entertainment, esthetics, and escapism, on brand attitude. In addition, the current study investigates how brand attitude affects brand loyalty. Lastly, this study explores the differences in the proposed model based on the type of service providers, such as robot servers and human servers.

## 2. Literature Review

### 2.1. The Role of Robots in the Hospitality Industry

Recently, robots have been used in various ways in the hospitality industry. For example, the Henna Hotel in Japan was the first hotel to employ robots in 2015 [2]. The robots employed at the Henna Hotel speak different languages, which could help the diverse customer base. In the beginning, a large number of robots were employed in the hotel, but after carefully reviewing customer feedback, managers adapted the ratio of human and robots by identifying exactly what the robots’ strengths were. This allowed the hotel operators to incorporate robot services while maximizing the benefits for the hotel. Another famous hotel robot is Connie, who works at Hilton Hotels and provides concierge services. Connie is a robot that can learn by interacting with customers [10]. This allows the robot to continue to learn how to provide customers with excellent service. These examples indicate that it is not a question of whether robots will be used in the service industry, it is merely a matter of when and how robots will be incorporated.

Not only are robots being seen in hotels, they are also being used in a variety of ways in restaurants. For example, Bionic Bar is a robot bartender who works for Royal Caribbean Cruises. The bartender robot has the ability to make two blended beverages in one minute, which can help customers receive their drinks in an efficient manner [11]. Haidilao, a hotpot restaurant in China, uses robots to operate an automated kitchen. Robot servers also bring the dishes to the customers while employing human staff to work with robots [12]. Even further advancements are being made, and a company in South Korea has introduced a delivery robot [13]. Customers can place a delivery order through a smartphone application, and robots can deliver even to apartment residences. Robots in the restaurant industry can assume various roles, offer services where they are most needed, and be incorporated into any establishment.

According to Chuah et al. [12], in the last 30 years, labor costs have increased by more than two times, and in the same time frame, the prices of robots have been reduced by half. It is also well known that restaurant employees may suffer injuries while performing duties because some jobs require working with, for example, sharp items and hot ovens and surfaces. Robots may help reduce workplace injury by performing more dangerous jobs. Lastly, it is necessary to mention that robot servers help restaurants offer contactless services, which customers desire in the current pandemic situation. Considering the many benefits robots can bring to the restaurant industry, it can be assumed that robots will increase in the future.

### 2.2. The Experience Economy

Consumers consider functional aspects, such as taste and portion size when eating at a restaurant, but they also value how much experiential benefits the restaurant provides [14]. The concept of experiential benefits has been studied for a long time because it provides customers with an unforgettable experience, which also significantly impacts their behavioral intentions [15]. Such experiential benefits can be conceptualized as “experience economy”, which was first suggested by Pine and Gilmore [16]. The experience economy includes four sub-dimensions: education, entertainment, esthetics, and escapism.

First, education in the experience economy means a desire to learn new things [16]. Humans become curious about new things, and they try to learn to satisfy their curiosity. For instance, when customers receive the services provided by the robot server, they are more likely to have an interest and curiosity about the robot, through which they would acquire new knowledge. Second, entertainment can be defined as the act of amusing or entertaining people [17]. Customers have unforgettable pleasant memories through entertainment, leading to positive behavioral intentions [15]. The restaurant industry is no exception. If robot servers provide services, various new experiences occurring in the service process will entertain customers. Third, esthetics refers to a customer’s interpretation of the physical environment surrounding them [18]. In particular, since the physical environment has an important influence on consumer emotions, efforts are needed to maximize aesthetic factors in the service industry [19]. For instance, if the robotic server were attractive, a customer would have a high level of esthetics. Fourth, escapism means that people removed themselves from their daily lives and look for new things [20]. People seek new experiences to escape from boring everyday life, and through such experiences, they can relax mentally and physically [21]. If customers receive robot services that they have not usually experienced in the restaurant industry, they will perceive a high level of escapism.

### 2.3. Effect of Experience Economy on Brand Attitude

Brand attitude refers to a consumer’s overall assessment of a brand [22]. It can be seen as a consumer’s attitude toward a certain brand evaluated based on its characteristics. Although customer satisfaction and brand attitude are used similarly, there is a clear difference between them. For instance, customer satisfaction is formed by a customer’s assessment of a specific transaction for a product/service [23], while brand attitude is formed through a comprehensive evaluation of various products released by the brand [24,25]. For this reason, brand attitude is known to be a greater concept than customer satisfaction [26]. In addition, it is important to understand the brand attitude, which helps establish a product positioning strategy [27].

This study proposes the effect of the experience economy on brand attitude based on the following theoretical and empirical backgrounds. When consumers use a certain product, their attitude toward the product depends on how many exceptional benefits they have received [18], which suggests that the experience economy can be an important predictor of brand attitude. For example, if restaurant customers have a high level of benefits while eating, they then positively affect the restaurant brand. Empirical research has also supported the relationship between experience economy and brand attitude. For instance, Park et al. [28] suggested that the experience economy plays an important role in forming satisfaction in the tourism industry. In addition, Mehmetoglu and Engen [29] found that when people have good experiences in tourist attractions, they perceive a high level of satisfaction. Loureiro [30] indicated that people have good feelings about a tourist destination when they have a good experience through tourism. More recently, Lee, Jeong, and Qu [31] showed that the experience economy is a key factor in making theme park visitors feel good. Therefore, this study proposes the following hypotheses.

**Hypothesis** **1** **(H1).***Education has a positive influence on brand attitude*.

**Hypothesis** **2** **(H2).***Entertainment has a positive influence on brand attitude*.

**Hypothesis** **3** **(H3).***Esthetics has a positive influence on brand attitude*.

**Hypothesis** **4** **(H4).***Escapism has a positive influence on brand attitude*.

### 2.4. Effect of Brand Attitude on Brand Loyalty

Brand loyalty is defined as “a deeply held psychological commitment to rebuy or repatronize a preferred product/service consistently in the future, thereby causing repetitive same brand or same brand-set purchasing, despite situational influences and marketing efforts having the potential to cause switching behavior” [32] (p. 34). Customers with a high level of loyalty to a certain brand devote themselves to the brand through continuous purchases in the future. In addition, many scholars have commonly suggested that loyal customers have the following characteristics [33,34,35]. First, they have an intention to spend extra money on the brand even though the price is higher than they expected. Second, loyal customers have a very high level of trust in a brand, so they consider the brand first when purchasing a product. Third, they are unlikely to switch to other brands. For these reasons, understanding the formation of brand loyalty is important to improve corporate performance.

In addition, this study proposes the relationship between brand attitude and brand loyalty based on the following theoretical and empirical backgrounds. According to the theory of planned behavior [36], individuals tend to perform actions concerning a certain object when they have a favorable attitude toward it. In addition, previous studies empirically found the effect of brand attitude on brand loyalty. For example, Oh and Park [37] found that the consumers’ brand attitude plays an important role in forming brand loyalty in the airline industry. Liu et al. [38] also suggested that brand attitude positively affects brand loyalty in the restaurant industry. More recently, Hwang, Choe et al. [7] revealed that when customers have a positive attitude toward a coffee brand, they are more likely to show a high level of brand loyalty. Therefore, this study proposes the following hypothesis.

**Hypothesis** **5** **(H5).***Brand attitude has a positive influence on brand loyalty*.

### 2.5. Difference Analysis according to the Type of Service Providers

As technology advances, robots are in charge of humans in many service fields. In the restaurant industry, robots also provide services to customers on behalf of humans, such as welcoming customers and taking their orders [8,39]. Although there is a negative view that these robot services take away human jobs [40], the positive impact of robot services on the restaurant industry is significant. For instance, the minimum wage has continuously increased in Korea, so labor costs are becoming a burden for restaurant companies. However, as robots take over human tasks, restaurant companies can relieve this burden [41]. In addition, since robots operate according to the input program, it has the advantage of minimizing mistakes in delivering services to customers, leading to high levels of satisfaction in these customers [42].

Empirical studies related to robot services have been conducted focusing on consumers. For example, Chiang and Trimi [43] suggested that customers have high satisfaction levels when they perceive the assurance and reliability of robotic services. In addition, Hwang et al. [7] found that customers tend to have memorable brand experiences when they perceive utilitarian and hedonic values from robots. Hwang et al. [44] also revealed that brand satisfaction is formed by four types of experiences: sensory, affective, behavioral, and intellectual brand experiences, in the context of robotic baristas. Based on the theoretical and empirical backgrounds discussed above, this study proposes the following hypotheses.

**Hypothesis** **6** **(H6).***There are significant differences in the mean based on the type of service providers*.

**Hypothesis** **6a** **(H6a).***There are significant differences in the mean for education based on the type of service providers*.

**Hypothesis** **6b** **(H6b).***There are significant differences in the mean for entertainment based on the type of service providers*.

**Hypothesis** **6c** **(H6c).***There are significant differences in the mean for esthetics based on the type of service providers*.

**Hypothesis** **6d** **(H6d).***There are significant differences in the mean for escapism based on the type of service providers*.

**Hypothesis** **6e** **(H6e).***There are significant differences in the mean for brand attitude based on the type of service providers*.

**Hypothesis** **6f** **(H6f).***There are significant differences in the mean for brand loyalty based on the type of service providers*.

### 2.6. Proposed Conceptual Model

Based on the six hypotheses, the following conceptual model is proposed in Figure 1.

## 3. Methodology

### 3.1. Measurement Items

The current study operationalizes multiple-measurement scales developed by previous studies to measure six constructs in the proposed model. First, to the experience economy, including education, entertainment, esthetics, and escapism, this study employed 12 items adopted from Hosany and Witham [18] and Hwang and Lee [5]. Second, the brand attitude was measured using three items drawn from Hwang and Hyun [45] and Mitchell and Olson [46]. Lastly, brand loyalty was measured using three items from Hwang and Park [47] and Zeithaml, Berry, and Parasuraman [48]. The questionnaire was finalized based on the measurement items mentioned above, and a seven-point Likert’s scale, which ranged from (1) strongly disagree to (7) strongly agree, was used to evaluate them.

### 3.2. Data Collection

To collect data, this study selected the M restaurant brand in Korea because the restaurant brand operates both robot and human server systems. In the case of robot servers, robots receive orders or provide food instead of humans (hereinafter RR). On the other hand, in terms of human servers, all services are performed by humans (hereinafter RH). Appendix A shows two types of restaurants. Since the same brand operates the two restaurants, only the service subjects are different, but different attributes such as food, physical environment, and price are the same. Although there is a slight difference in the layout of the physical environment between the two restaurants, this part does not significantly affect the results of this study. The survey was conducted at each of the two restaurants.

This study collected data using Company E, one of the largest data companies in South Korea. Based on the rules of Korea Centers for Disease Control and Prevention (KCDCP), it is allowed to conduct one-on-one interviews on the street. The company fully trained 10 interviewers in order to perform the survey. The interviewers waited at the restaurant entrance, and posed questions to customers who had finished eating at the restaurant. First, the respondents were asked whether they used the restaurant or not, and if they did not use the restaurant, they were excluded from the survey. The purpose of the survey was fully explained to the respondents before the survey started, and after the survey was finished, a gift of about USD 5 was presented to the respondents as a token of appreciation.

After deleting four multivariate outliers, 296 responses were used for statistical analysis regarding RR. In terms of RH, after removing 34 multivariate outliers, 294 responses were employed for statistical analysis.

## 4. Data Analysis

### 4.1. Profile of the Respondents

Table 1 presents the sociodemographic characteristics of the respondents. First, in terms of RR, 41.9% (*n* = 124) were male, while 58.1% (*n* = 172) were female. Among the respondents, those in their 30s were the highest in number (*n* = 107 and 36.1%), followed by those in their 40s (*n* = 79 and 26.7%). In addition, 82% of the respondents (*n* = 245) had a bachelor’s degree, and 55.4% of the respondents (*n* = 164) were married. Lastly, the highest percentage of respondents at 28.4% (*n* = 84) earned a monthly income between USD 4001 and USD 5000.

Second, in the case of RH, the proportion of respondents between males and females was the same. The number of respondents in their 30s was the largest group (*n* = 97 and 33.3%). Regarding the education level, the majority of the respondents graduated from college (*n* = 187 and 63.6%). In addition, 54.1% of the respondents (*n* = 159) were married, and 26.2% of the respondents (*n* = 77) earned a monthly income level between USD 3001 and USD 4000.

### 4.2. Confirmatory Factor Analysis

Table 2 shows the variables used in this study with their standardized factor loadings. In addition, the two data, RR and RH, were merged (hereinafter MTD). The results of the confirmatory factor analysis (CFA) indicated an appropriate model fit for all three models (RR: χ^2^ = 249.188, df = 120, χ^2^/df = 2.077, *p* < 0.001, NFI = 0.951, CFI = 0.974, TLI = 0.967, and RMSEA = 0.061; RH: χ^2^ = 316.589, df = 120, χ^2^/df = 2.188, *p* < 0.001, NFI = 0.937, CFI = 0.959, TLI = 0.948, and RMSEA = 0.074; and MTD: χ2 = 465.806, df = 120, χ^2^/df = 3.882, *p* < 0.001, NFI = 0.954, CFI = 0.965, TLI = 0.956, and RMSEA = 0.070) [49]. All of the factor loading values were higher than or equal to 0.780 for the RR model, 0.830 for the RH model, and 0.849 for the MTD model.

As presented in Table 3, the average variance extracted (AVE) values for all three of the models were greater than 0.50, suggesting an acceptable convergent validity [50]. The composite reliabilities of all of the constructs were greater than 0.70, indicating a satisfactory level of internal consistency [51]. Lastly, the AVE values were higher than the squared correlation (R2) values between each pair of constructs, which supported a high level of discriminant validity [52].

### 4.3. Structural Equation Modeling

Figure 2 shows the results of the structural equation modeling (SEM) analysis, and the results revealed that the three models had an appropriate fit (RR: χ^2^ = 339.823, df = 124, χ^2^/df = 2.741, *p* < 0.001, NFI = 0.934, CFI = 0.957, TLI = 0.946, and RMSEA = 0.078; RH: χ^2^ = 447.945, df = 124, χ^2^/df = 3.612, *p* < 0.001, NFI = 0.906, CFI = 0.930, TLI = 0.914, and RMSEA = 0.094; and MTD: χ^2^ = 641.782, df = 124, χ^2^/df = 5.176, *p* < 0.001, NFI = 0.935, CFI = 0.947, TLI = 0.935, and RMSEA = 0.084). The data analysis indicated that education positively affects brand attitude in RR; however, there is no relationship between the two concepts in RH and MTD. Therefore, Hypothesis 1 was partially accepted. Second, entertainment has a positive influence on brand attitude in RR, RH, and MTD, so Hypothesis 2 was accepted. Third, the effect of esthetics on brand attitude was identified, except for RR. Hence, Hypothesis 3 was partially accepted. Fourth, escapism plays an important role in the formation of brand attitude in all three models, so Hypothesis 4 was accepted. Lastly, the relationship between brand attitude and brand loyalty was found in RR, RH, and MTD, Thus, Hypothesis 5 was accepted.

### 4.4. Results of t-Tests

As presented in Table 4, the *t*-tests were conducted to evaluate the differences of average values among the six concepts, such as education, entertainment, esthetics, escapism, brand attitude, and brand loyalty based on the type of service providers, including robots and humans. The results of the data analysis indicated significant differences in average values in all six of the concepts based on the type of service providers. Thus, Hypothesis 6 was supported. In particular, the average values were greater in RR than in RH for all six concepts.

## 5. Discussion and Implications

This study was designed to investigate the experience economy in the context of the robotic restaurant since the importance of the robot’s role in the hospitality industry. More specifically, it was proposed that the experience economy positively affects the brand attitude. In addition, the brand attitude was hypothesized to influence brand loyalty. Lastly, the differences in the type of service providers were proposed during the theory-building process. This study employed 296 robot-serving restaurant customers and 294 human-serving restaurant customers to evaluate the six hypotheses. The results provide the following theoretical and managerial implications.

### 5.1. Theoretical Implications

First, this study applied the concept of the experience economy to the restaurant industry and presented a theoretical extension that proves the causal relationship between the experience economy and brand attitude. As experiential benefits become more important in restaurants, this study interpreted the phenomenon as “experience economy” [14,15,16]. Additionally, based on the review of previous studies, the four sub-dimensions of the experience economy were applied to human-serving restaurants and robot-serving restaurants [15,16,18,19,20,21]. Furthermore, focusing on previous research in the tourism industry, this study proposed a hypothesis that in the restaurant industry, the experience economy would have a positive effect on brand attitude [18,28,29,30,31]. The analysis results for the hypothesis testing are as follows. In robot-serving restaurants, three sub-dimensions of education, entertainment, and escapism positively affected brand attitude. In human-serving restaurants, three sub-dimensions of entertainment, esthetics, and escapism showed a statistically positive effect on brand attitude. This study presented the results that prove the causal relationship between the experience economy and brand attitude in the restaurant industry for the first time.

Second, this study proved the positive effect of brand attitude on brand loyalty in the restaurant industry. Liu et al. [38] and Hwang et al. [7] also proved the causal relationship between brand attitude and brand loyalty in the restaurant industry. Still, this study has a differential theoretical implication compared to these two studies. This study focused on the social background in which robot services were applied in the restaurant industry [8,39]. Moreover, the causal relationship was proven by distinguishing robot-serving restaurants and human-serving restaurants. The hypotheses testing results confirmed that brand attitude positively affects brand loyalty in both types. Furthermore, the human-serving restaurants showed a higher path coefficient than robot-serving restaurants. Accordingly, this study presented a theoretical extension by proving the causal relationship between brand attitude and brand loyalty according to robot and human.

Third, this study covered the differences for each type of service provider (i.e., human-serving versus robot-serving). In accordance with technological advancement, robot servers have appeared in the foodservice industry, and various media have presented positive and negative views about it [8,39,40,41,42]. In this vein, this research reviewed previous studies and proposed that there will be differences in the averaged customer responses by the types of service providers [7,43,44]. In addition, while verifying each causal relationship, the analysis results for the robot server, human server, and overall merged types were presented. The analysis results regarding the differences for each service provider type revealed statistically significant differences in all seven variables. Furthermore, the robot restaurant showed a higher value in all variables in terms of the averaged responses. Therefore, this study presented theoretical implications that prove a difference in customer response for each type of service provider.

Lastly, the findings revealed that the causal relationship between the experience economy and brand attitude, the esthetics variable in the robot-serving restaurants, and the education variable in the human-serving restaurants were not statistically significant. The esthetic variable was measured by server appearance and attractiveness. Regarding the human servers, their appearance and attractiveness may influence the brand attitude towards the restaurant. However, the robots may not affect the brand attitude because robots, which have a monolithic appearance and are engrossed in the given functional performance. Education was measured by learning experience or curiosity. When consumers meet robot servers, they will recognize the technological development of the period through a novel experience. In this vein, consumers may perceive the experience economy of education and form a positive brand attitude in the robot-serving restaurant. However, encountering human servers is a mundane experience for consumers. It may not cause learning experiences or curiosity, and even if the consumers experienced such things, the impact on brand attitude would have been negligible. This study shows that the different sub-dimensions of the experience economy were rejected for each type of service provider. Accordingly, this study is the first to present results revealing that the experience economy differs by the type of service provider in brand attitude formation.

### 5.2. Managerial Implications

This study found that brand attitude enhances brand loyalty in both RR and RH. Managers and marketers should focus more on consumers’ brand attitudes to improve brand loyalty. In RR, education, entertainment, and escapism aid in increasing brand attitude. In addition, in RH, entertainment, esthetics, and escapism help to enhance brand attitude. Thus, the results suggest practical strategies for each restaurant type as follows.

Entertainment is the most important factor in forming brand attitude in RR. As described earlier, entertainment can be defined as the act of amusing or entertaining people. Zhang et al. [53] suggested robots should be augmented to have the ability to make humorous responses and use witty language in order to improve entertainment services. That is, robots should be able to engage in humorous language when interacting with consumers rather than just a typical service language. For instance, when consumers order spicy food, the robot can say, “Welcome to the spicy food challenge! Hey buddy, you should be ready to call 911” instead of just taking orders. In addition, escapism also aids in the enhancement of brand attitude in RR. Marketers for RR should provide escapism experiences due to consumer stress brought on by the COVID-19 pandemic. Consumers prefer non-face-to-face services to prevent infection after the outbreak of the pandemic [9]. For instance, marketers can entice consumers with messages such as “Enjoy your gourmet life without worrying about the pandemic” and provide completely non-face-to-face services by robots. By recognizing these advantages, consumers would have a comfortable experience as if they have escaped from social distancing life when they visit RR. Lastly, education is also a significant variable in forming brand attitudes in RR. Therefore, robot servers should provide more educational experiences. Sauppé and Mutlu [54] note that the robot’s screen, which doubles as a face, can offer more information. For instance, the robot servers can efficiently inform consumers about culinary processes and menu descriptions on their screen when serving menus. These robot service experience strategies will aid in the enhancement of RR’s brand attitude and improve brand loyalty.

Moreover, in the RH, entertainment was the most significant factor in affecting brand attitude. Thus, RH requires creating entertainment service strategies differently when compared with RR. For example, Etude House, which is a cosmetic brand, welcomes consumers with its interesting concept and princess aesthetic [55]. More specifically, the service concept is that the employees are the princesses of Etude Kingdom and consumers are neighboring countries’ princesses, thus they call consumers “Princesses”. In other words, the unique service concept provides entertainment services with a fun conversation manual tailored to the brand. For instance, managers can compose a service manual with a dialogue that appropriately mixes either dialect or language of the country when serving cuisine from the specific country. In addition, human servers’ esthetics had a positive effect on brand attitude toward RH, and it was not supported by RR. Despite the issue of the burden of esthetics on employees [56], this is still an important element in hospitality industries. Managers should consider refined uniforms and create a service manual for elegant gestures and service behaviors, excluding personal esthetic elements such as personal grooming, hairstyle, and hair length. Lastly, escapism is also a significant variable in forming brand attitudes in RH. Consumers want to have new experiences which make them feel they have escaped from their daily routines [57]. Managers should design serving processes for a special experience that can only be provided by professional human servers, such as flambé and bartending performances on consumers’ tables. These experience strategies would improve brand attitude toward RH and aid in the enhancement of consumers’ loyalty.

## 6. Limitations and Future Research

This study has the following limitations. First, since this study collected data only from the M restaurant brand in South Korea, it is somewhat difficult to generalize. Further research is needed to generalize the study with samples collected from various restaurant brands in other countries. Second, this study focused on face-to-face services such as robot servers in the restaurant industry, so it is recommended to study non-face-to-face services such as robot chefs in future studies. Third, the data analysis results showed that the average values were higher in RR than in RH for all six concepts: education, entertainment, esthetics, escapism, brand attitude, and brand loyalty. These results can be attributed to the tendency of consumers to prefer non-face-to-face services in the current COVID-19 situation. Therefore, in future studies, it is necessary to compare robot servers and human servers based on the proposed model of this study after the end of COVID-19.

## Figures and Tables

**Figure 1 ijerph-19-03430-f001:**
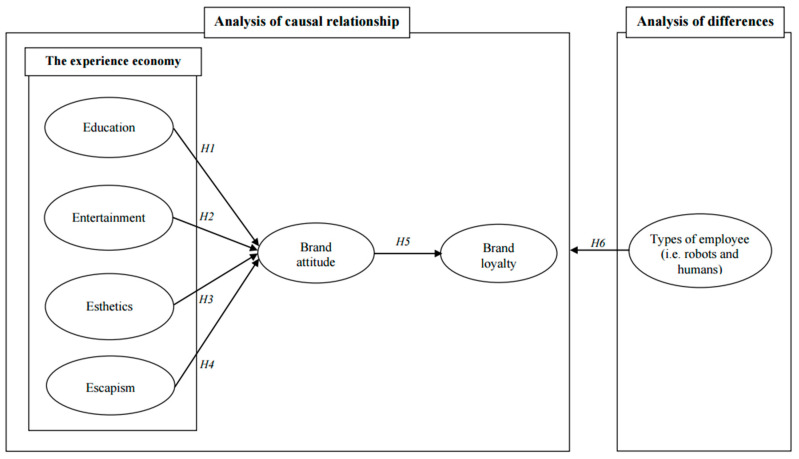
Proposed conceptual model. Notes: H = hypothesis.

**Figure 2 ijerph-19-03430-f002:**
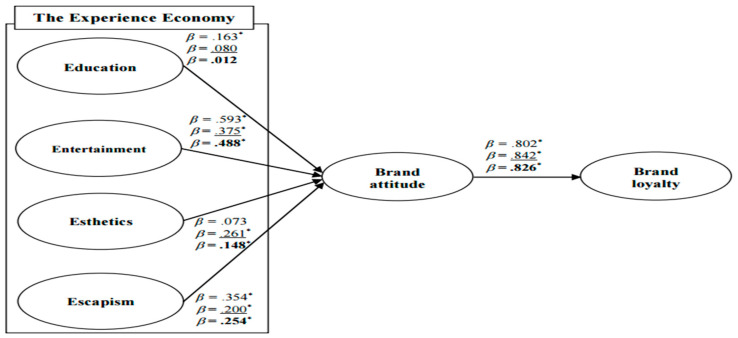
Standardized theoretical path coefficients. RR: χ^2^ = 339.823, df = 124, χ^2^/df = 2.741, *p* < 0.001, NFI = 0.934, CFI = 0.957, TLI = 0.946 and RMSEA = 0.078. RH: χ^2^ = 447.945, df = 124, χ^2^/df = 3.612, *p* < 0.001, NFI = 0.906, CFI = 0.930, TLI = 0.914 and RMSEA = 0.094. MTD: χ^2^ = 641.782, df = 124, χ^2^/df = 5.176, *p* < 0.001, NFI = 0.935, CFI = 0.947, TLI = 0.935 and RMSEA = 0.084. Notes 1: RR = a restaurant where robots provide services, RH = a restaurant where humans provide services and MTD = Merging two data. Notes 2: The unmarked values are for RR; the underlined values are for RH; and values in boldface type are for MTD. Notes 3: NFI = normed fit index, IFI = incremental fit index, CFI = comparative fit index, TLI = Tucker–Lewis index, and RMSEA = root mean square error of approximation. Notes 4: * *p* < 0.05.

**Table 1 ijerph-19-03430-t001:** Profile of survey respondents.

Variable	A Restaurant Where Robots Provide Services (*n* = 296)	A Restaurant Where Humans Provide Services (*n* = 294)	Merging Two Data (*n* = 590)
**Gender**			
Male	124 (41.9%)	147 (50.0%)	271 (45.9%)
Female	172 (58.1%)	147 (50.0%)	319 (54.1%)
**Age**			
20s	63 (21.3%)	70 (23.8%)	133 (22.5%)
30s	107 (36.1%)	97 (33.0%)	204 (34.6%)
40s	79 (26.7%)	70 (23.8%)	149 (25.3%)
50s	47 (15.9%)	57 (19.4%)	104 (17.6%)
**Education level**			
Less than high school diploma	9 (3.0%)	20 (6.8%)	29 (4.9%)
Associate’s degree	25 (8.4%)	36 (12.2%)	61 (10.3%)
Bachelor’s degree	245 (82.8%)	187 (63.6%)	423 (73.2%)
Graduate degree	17 (5.7%)	17.3 (51%)	68 (11.5%)
**Marital status**			
Single	131 (44.3%)	134 (45.6%)	265 (44.9%)
Married	164 (55.4%)	159 (54.1%)	323 (54.7%)
Others	1 (.3%)	1 (0.3%)	2 (0.3%)
**Income level**			
USD 6001 and over	76 (25.7%)	60 (20.4%)	136 (23.1%)
USD 5001–6000	67 (22.6%)	47 (16.0%)	114 (19.3%)
USD 4001–5000	84 (28.4%)	47 (16.0%)	131 (22.2%)
USD 3001–4000	46 (15.5%)	77 (26.2%)	123 (20.8%)
USD 2001–3000	13 (4.4%)	41 (13.9%)	54 (9.2%)
Under USD 2000	10 (3.4%)	22 (7.5%)	32 (5.4%)

**Table 2 ijerph-19-03430-t002:** Confirmatory factor analysis: items and loadings.

Construct and Scale Item	Standardized Loading ^a^		
	RR	RP	MTD
**Experience economy**			
**Education**			
This robotic server made me more knowledgeable.	0.855	0.896	0.898
This robotic server stimulated my curiosity to learn new things.	0.912	0.944	0.946
This robotic server provided a real learning experience.	0.780	0.883	0.871
**Entertainment**			
This robotic server kept me amused.	0.903	0.902	0.902
This robotic server was entertaining.	0.891	0.901	0.898
This robotic server was fun.	0.943	0.895	0.917
**Esthetics**			
The appearance of the robotic server was good.	0.865	0.883	0.869
The robot server looked good.	0.895	0.921	0.902
The robotic server was attractive.	0.929	0.834	0.884
**Escapism**			
I felt I was in a different world while using this robot server.	0.900	0.847	0.884
I completely escaped from my daily routine while the robotic server offered me its services at this restaurant.	0.787	0.884	0.849
I felt like I was in a different place while using this robot server at this restaurant.	0.906	0.891	0.906
**Brand attitude**			
Unfavorable–Favorable	0.861	0.830	0.881
Negative–Positive	0.832	0.917	0.898
Bad–Good	0.911	0.923	0.927
**Brand loyalty**			
I say positive things about this restaurant brand to others.	0.895	0.899	0.900
I would like to use this restaurant brand more often.	0.873	0.855	0.866
I would like to use this restaurant brand in the future.	0.909	0.873	0.882

Notes 1: RR = a restaurant where robots provide services, RH = a restaurant where humans provide services, and MTD = merging two data. Notes 2: ^a^ All factor loadings are significant at *p* < 0.001. Notes 3: NFI = normed fit index, IFI = incremental fit index, CFI = comparative fit index, TLI = Tucker–Lewis index, and RMSEA = root mean square error of approximation.

**Table 3 ijerph-19-03430-t003:** Descriptive statistics and associated measures.

	Mean (Std Dev.)	AVE	(1)	(2)	(3)	(4)	(5)	(6)
(1) Education	5.31 (1.05) 4.30 (1.27) **4.81 (1.27)**	0.724 0.825 **0.820**	0.887 0.934 **0.932**	0.731 ^a^ 0.459 **0.563**	0.654 0.382 **0.489**	0.766 0.611 **0.696**	0.528 0.313 **0.409**	0.675 0.523 **0.583**
(2) Entertainment	5.70 (0.82) 5.46 (0.90) **5.58 (0.87)**	0.833 0.809 **0.821**	0.534 ^b^ 0.211 **0.317**	0.937 0.927 ** 0.933 **	0.768 0.804 **0.794**	0.767 0.690 **0.718**	0.742 0.430 **0.534**	0.812 0.830 **0.827**
(3) Esthetics	5.65 (0.90) 5.48 (0.90) **5.56 (0.90)**	0.804 0.774 **0.783**	0.428 0.146 **0.239**	0.590 0.646 **0.630**	0.925 0.911 ** 0.783 **	0.798 0.704 **0.740**	0.650 0.423 **0.499**	0.720 0.721 **0.730**
(4) Escapism	5.57 (0.85) 5.10 (1.03) **5.34 (0.97)**	0.750 0.764 **0.774**	0.587 0.373 **0.484**	0.588 0.476 **0.516**	0.637 0.496 **0.548**	0.900 0.907 ** 0.774 **	0.688 0.449 **0.544**	0.750 0.672 **0.706**
(5) Brand attitude	5.87 (0.74) 5.69 (1.05) **5.78 (0.91)**	0.754 0.794 **0.814**	0.279 0.098 **0.167**	0.551 0.185 **0.285**	0.423 0.179 **0.249**	0.473 0.202 **0.296**	0.902 0.920 ** 0.814 **	0.733 0.433 **0.531**
(6) Brand loyalty	5.62 (0.82) 5.35 (1.00) **5.49 (0.92)**	0.796 0.767 **0.779**	0.456 0.274 **0.340**	0.659 0.689 **0.684**	0.518 0.520 **0.533**	0.563 0.452 **0.498**	0.537 0.187 **0.282**	0.921 0.908 ** 0.779 **

Notes 1: The unmarked values are for a restaurant where robots provide services; the underlined values are for a restaurant where humans provide services; and values in boldface type are for merging two data. Notes 2: AVE = average variance extracted. Notes 3: Shaded numbers, composite reliabilities are along the diagonal. Notes 4: ^a^ correlations are above the diagonal and ^b^ squared correlations are below the diagonal.

**Table 4 ijerph-19-03430-t004:** Results of *t*-tests: Effects of types of service providers on experience economy, brand attitude, and brand loyalty.

		Types of Service Providers			
		A Restaurant Where Robots Provide Services	A Restaurant Where Humans Provide Services	*t*-Value	*p*-Value
Experience economy	Education	5.31	4.30	10.516	0.000 *****
	Entertainment	5.70	5.46	3.428	0.001 ****
	Esthetics	5.65	5.48	2.268	0.024 ***
	Escapism	5.57	5.10	6.037	0.000 *****
		A restaurant where robots provide services	A restaurant where humans provide services	*t*-value	*p*-value
Attitude		5.87	5.69	2.433	0.015 ***
		A restaurant where robots provide services	A restaurant where humans provide services	*t*-value	*p*-value
Brand loyalty		5.62	5.35	3.564	0.000 *****

Notes: * *p* < 0.05, ** *p* < 0.01, *** *p* < 0.001.

## Data Availability

Data sharing is not applicable.

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
