# Peer review of "The Effects of Types of Service Providers on Experience Economy, Brand Attitude, and Brand Loyalty in the Restaurant Industry"

_ijerph, 2022, doi:10.3390/ijerph19063430_

Round 1

Reviewer 1 Report

The topic is interesting with the comparison of the human and robots as the employees. In addition, the author used the experience economy dimensions to determine the brand attitude. However, the author needed to add more details regarding the data collection process and the screening process for the respondents. The general information about the restaurant should also be added, including similarity or differences in the restaurant services between humans and robots. The results and discussions were presented sufficiently and easy to understand. Under Figure 2 in the note, the author used the term "coffee shop". Is it the coffee shop or restaurant? Please be consistent. The managerial implications for all the results should be clearly explained, including for brand attitude and brand loyalty. This is to offer the actionable recommendations for the practitioners, especially how to improve brand attitude and brand loyalty. This should be one of the most important parts of this study. The limitation of one restaurant should be addressed it more clearly. The author should also explain the generalization of the study, being beneficial to future research.

Author Response

Reviewer 1

  1. The topic is interesting with the comparison of the human and robots as the employees. In addition, the author used the experience economy dimensions to determine the brand attitude.

Response: Thank you for this encouragement and for all of your comments. Our responses to your comments are summarized in the following section.

  1. However, the author needed to add more details regarding the data collection process and the screening process for the respondents.

Response: Thank you for this keen observation. We added the details about data collection procedure based on the Korea Centers for Disease Control and Prevention (KCDCP) rules in the revised manuscript. Please see the methodology section of the revised manuscript.

☞ This study collected data using Company E, one of the largest data companies in South Korea. Base on the rules of Korea Centers for Disease Control and Prevention (KCDCP), it is allowed to conduct one-on-one interviews on the street. The company fully trained 10 interviewers in order to perform the survey. The interviewers waited at the restaurant entrance, and asked customers who have finished eating at the restaurant. First, the respondents were asked whether they use the restaurant or not, and if they did not use the restaurant, they were excluded from the survey. The purpose of the survey was fully explained to the respondents before the survey started, and after the survey was finished, a gift of about US$5 was presented to the respondents as a token of appreciation.

  1. The general information about the restaurant should also be added, including similarity or differences in the restaurant services between humans and robots.

Response: Thank you for this comment. Before collecting data, our research team thought a lot about choosing a restaurant brand. As a result, we selected ‘Madforgarlic,’ a Korean restaurant brand. The reason is that, as we explained, Madforgarlic operates both RR (robots receive orders or provide food instead of humans) and RH (all services are performed by humans) and in particular, the brand is a chain, so each branch has similar characteristics. Prior to the data collection, we confirmed through a preliminary survey that all branches offer the same price of menus, and the quality of food is not significantly different. However, although the physical environment was almost similar, it was confirmed that there was a slight difference in layout. However, we decided to collect data from Madforgarlic because we thought only this part did not affect the results of this study significantly. However, as you mentioned, we added the following sentence in order to explain this issue.

☞ Since the same brand operates the two restaurants, only the service subjects are different, but different attributes such as food, physical environment, and price are the same. Although there is a slight difference in the layout of the physical environment between the two restaurants, this part does not significantly affect the results of this study.

  1. The results and discussions were presented sufficiently and easy to understand.

Response: We thank the reviewer for this positive assessment.

  1. Under Figure 2 in the note, the author used the term "coffee shop". Is it the coffee shop or restaurant? Please be consistent.

Response: Thank you for this comment, and we revised the typo.

  1. The managerial implications for all the results should be clearly explained, including for brand attitude and brand loyalty. This is to offer the actionable recommendations for the practitioners, especially how to improve brand attitude and brand loyalty. This should be one of the most important parts of this study.

Response: Thank you for the important comment, and we rewrote managerial implications in order to clearly explain practical strategies for brand attitude and brand loyalty. In addition, we provided specific practical implications with examples for each type of restaurant (i.e. RR and RH) for restaurant managers.

☞ This study found that brand attitude enhances brand loyalty in both RR and RH. Managers and marketers should focus more on consumers’ brand attitudes to improve brand loyalty. In RR, education, entertainment, and escapism aid to increase brand attitude. In addition, in RH, entertainment, esthetics, and escapism help to enhance brand attitude. Thus, the results suggest practical strategies for each restaurant type as follows.

Entertainment is the most important factor in forming brand attitude in RR. As described earlier, entertainment can be defined as the act of amusing or entertaining people. Zhang, Gursoy, and Shi [53] suggested robots should be augmented to have the ability to make humorous responses and use witty language in order to improve entertainment services. That is, robots should be able to engage in humorous language when interacting with consumers rather than just a typical service language. For instance, when consumers order spicy food, the robot can say, “Welcome to the spicy food challenge! Hey buddy, you should be ready to call 911” instead of just taking orders. In addition, escapism also aids to enhance brand attitude in RR. Marketers at RR should provide escapism experiences as consumers have a lot of stress due to COVID-19. Consumers prefer non-face-to-face services to prevent infection after the outbreak of the pandemic [9]. For instance, marketers can entice consumers with messages such as “Enjoy your gourmet life without worrying about the pandemic” and provide completely non-face-to-face services by robots. By recognizing these advantages, consumers would have a comfortable experience as if they have escaped from social distancing life when they visit RR. Lastly, education is also a significant variable in forming brand attitudes in RR. Therefore, robot servers should provide more educational experiences. Sauppé and Mutlu [54] said the robot’s screen, which doubles as a face, can offer more information. For instance, the robot servers can efficiently inform consumers about culinary processes and menu descriptions on their screen when serving menus. These robot service experience strategies will aid to enhance RR's brand attitude and improve brand loyalty.

Also, in the RH, entertainment was the most significant factor in affecting brand attitude. Thus, RH requires making entertainment service strategies differently when compared with RR. For example, Etude House, which is a cosmetic brand, welcomes consumers with its interesting concept and princess aesthetic [55]. More specifically, the service concept is that the employees are the princesses of Etude Kingdom and consumers are neighboring countries’ princesses, thus they call consumers “Princess”. In other words, the unique service concept provides entertainment services with a fun conversation manual tailored to the brand. For instance, managers can compose a service manual with a dialogue that appropriately mixes either dialect or language of the country when serving cuisine from the specific country. In addition, human servers’ esthetics had a positive effect on brand attitude toward RH, and it was not supported by RR. Despite the issue of the burden of esthetic on employees [56], this is still an important element in hospitality industries. Managers should consider refined uniforms and create a service manual for elegant gestures and service behaviors, excluding personal esthetic elements such as personal grooming, hairstyle, and hair length. Lastly, escapism is also a significant variable in forming brand attitudes in RH. Consumers want to have new experiences which make them feel they have escaped from their daily routines [57]. Managers should design serving processes for a special experience that can only be provided by professional human servers, such as flambé and bartending performances on consumers' tables. These experience strategies would improve brand attitude toward RH and aid to enhance consumers’ loyalty.

  1. The limitation of one restaurant should be addressed it more clearly. The author should also explain the generalization of the study, being beneficial to future research.

Response: Thank you for your comment. As you suggested, we added the following sentences:

☞ First, since this study collected data only from the M restaurant brand in South Korea, it is somewhat difficult to generalize. Further research needs to generalize the study with samples collected from various restaurant brands in other countries.

Thank you so much for your comments and the time you gave for the improvement of our manuscript.

Reviewer 2 Report

Dear Authors,

Your research covers interesting issues, but the methodological side still needs a lot of work. The title of the article raises many doubts because it is methodologically incorrect. Robots have no legal personality and I have never before read about technological devices being referred to as employees. Robots should not be considered as a type of employee. The proposed title should be changes. It misleads the reader with the topic. What is more, these are too many problems which could not be thoroughly reviewed.

In my opinion, the main aim of the research (“to understand the relationships among the experience economy, brand attitude, and brand loyalty based on the type of employees, such as robot servers and human servers in the restaurant industry”) is not clear and is methodologically incorrect.  The objective is very general, too general for a single article in which such complex interrelationships could be exhaustively explained.

The research is very poorly rooted in the literature. The article refers to so many issues that none of them are satisfactorily described. The most controversial for me is the methodology. I had the impression that the form prevails over the content, and complicated mathematical calculations come to the fore in the analysis, which completely defeats the purpose of the study. You seemed to get lost in the hypotheses: e.g.  hypothesis 1 is identical to hypothesis 4. The number of hypotheses and proposals for various correlations is simply astonishing.

Such an approach is well known in social studies and is called P-fishing. It refers to the selective reporting of statistical results that cross some boundary of statistical significance. The problem arises because researchers may run many more analyses than they actually report, and given model or specification uncertainty, it is easy to report only those results that are consistent with the inferences that the analyst wants to make. 

The hypotheses are obvious and the statistical analyses do not add anything new. However, the complex statistical calculations do not hide many errors. I haven't find detailed information about the survey itself. After reading the article, I have no knowledge on how the survey questionnaire was constructed, how many questions it had, what the response scales were, how the sample was selected (I do not know why you decided on such a sample size?) etc. There is no information in the article about the ethics committee's approval of the research or where to check the research data. I have serious doubts that this is a ghost study.

I suggest to rewritten the paper and to analyze selected problems, not just to immerse in the statistical issues, but remember about the main question. 

Kind regards

Author Response

Reviewer 2

  1. Your research covers interesting issues, but the methodological side still needs a lot of work. The title of the article raises many doubts because it is methodologically incorrect. Robots have no legal personality and I have never before read about technological devices being referred to as employees. Robots should not be considered as a type of employee. The proposed title should be changes. It misleads the reader with the topic.

Response: Thank you for your comments. First, in terms of methodological issues, please refer to our response for your third comment. Second, we agree with your option that robots should not be considered as a type of employee. Thus, we changed the title of our paper as follows:

☞ The effects of types of service providers on experience economy, brand attitude, and brand loyalty in the restaurant industry

  1. In my opinion, the main aim of the research (“to understand the relationships among the experience economy, brand attitude, and brand loyalty based on the type of employees, such as robot servers and human servers in the restaurant industry”) is not clear and is methodologically incorrect. The objective is very general, too general for a single article in which such complex interrelationships could be exhaustively explained.

Response: Thank you for this critical comment. As you suggested, we tried to clearly explain main aim of the research and the method to attain the main aim in the revised manuscript.

☞ In summary, consumer experiences are an important factor in predicting consumer behavior in the service industry [5,6,7], but there are not many studies on it in background robotic restaurants. Therefore, this study aims to understand the importance of the experience economy and the effects of its outcome variables in robot restaurants. In addition, since the consumer experiences may vary depending on the service provider [8], this study attempts to find out the difference between robot servers and human servers in the experience economy for the first time in the restaurant industry. Specifically, this study examines the effect of four sub-dimensions of the experience economy, namely education, entertainment, esthetics, and escapism, on brand attitude. In addition, the current study investigates how brand attitude affects brand loyalty. Lastly, this study explores the differences in the proposed mod-el based on the type of service providers, such as robot servers and human servers.

References

  1. Hwang, J.; Lee, J. A strategy for enhancing senior tourists’ well-being perception: Focusing on the experience economy. Journal of Travel & Tourism Marketing2019, 36(3), 314-329.
  2. Park, M.; Oh, H.; Park, J. Measuring the experience economy of film festival participants. International Journal of Tourism Sciences2010, 10(2), 35-54.
  3. Hwang, J.; Choe, J.Y.J.; Kim, H.M.; Kim, J.J. Human baristas and robot baristas: How does brand experience affect brand satisfaction, brand attitude, brand attachment, and brand loyalty?. International Journal of Hospitality Management2021a, 99, 103050.
  4. Byrd, K.; Fan, A.; Her, E.; Liu, Y.; Almanza, B.; Leitch, S. Robot vs human: expectations, performances and gaps in off-premise restaurant service modes. International Journal of Contemporary Hospitality Management. 2021, 33(1), 3996-4016.

  1. The research is very poorly rooted in the literature. The article refers to so many issues that none of them are satisfactorily described. The most controversial for me is the methodology. I had the impression that the form prevails over the content, and complicated mathematical calculations come to the fore in the analysis, which completely defeats the purpose of the study. You seemed to get lost in the hypotheses: e.g. hypothesis 1 is identical to hypothesis 4. The number of hypotheses and proposals for various correlations is simply astonishing.

Response: Thank you for your important comment. In this study, there are two types of studies. The first study was designed to find the relationships among education, entertainment, esthetics, escapism, brand attitude, and brand loyalty. In addition, for this objective, we employed structural equation modeling in the AMOS program. The second study was designed to identify the differences between the types of service providers (i.e. robots and humans) in the experience economy, and the difference analysis was employed for this study. In the manuscript, all the data analysis results are presented in Figure 2 and Table 4. Lastly, we modified the typo in Hypothesis 4.

  1. Such an approach is well known in social studies and is called P-fishing. It refers to the selective reporting of statistical results that cross some boundary of statistical significance. The problem arises because researchers may run many more analyses than they actually report, and given model or specification uncertainty, it is easy to report only those results that are consistent with the inferences that the analyst wants to make.

Response: Thank you this comment. In this study, statistical significance was determined by the p value .05 widely used as a criterion for accepting hypotheses in social science. Furthermore, this study was conducted based on deductive research methods. That is, hypotheses were derived based on the existing theory and the hypotheses were verified using samples.

  1. The hypotheses are obvious and the statistical analyses do not add anything new. However, the complex statistical calculations do not hide many errors. I haven't find detailed information about the survey itself. After reading the article, I have no knowledge on how the survey questionnaire was constructed, how many questions it had, what the response scales were, how the sample was selected (I do not know why you decided on such a sample size?) etc. There is no information in the article about the ethics committee's approval of the research or where to check the research data. I have serious doubts that this is a ghost study.

Response: Thank you for these significant comments. First, we added the details about data collection procedure as follows:

☞ This study collected data using Company E, one of the largest data companies in South Korea. Base on the rules of Korea Centers for Disease Control and Prevention (KCDCP), it is allowed to conduct one-on-one interviews on the street. The company fully trained 10 interviewers in order to perform the survey. The interviewers waited at the restaurant entrance, and asked customers who have finished eating at the restaurant. First, the respondents were asked whether they use the restaurant or not, and if they did not use the restaurant, they were excluded from the survey. The purpose of the survey was fully explained to the respondents before the survey started, and after the survey was finished, a gift of about US$5 was presented to the respondents as a token of appreciation.

Second, in terms of sample size, this study used 296 responses for statistical analysis regarding RR. In terms of RH, 294 responses were employed for statistical analysis. The sample size in this study (n = 210) follows Hair et al.’s (1998) recommended size (greater than or equal to 200) when using CFA/SEM with the maximum likelihood estimation method. Considering the number of independent variables and ability of detecting R2 values in this research, the sample size in the present study was sufficient to validate the generalizability of the results (Hair et al., 1998). Accordingly, while we somewhat agree that our sample size for the group comparison is not large enough, this sample size wouldn’t much jeopardize to achieve the objectives of our study.

Lastly, there is no need for IRB approval because the data in this study are surveys, not respondents’ physical experiments. In addition, in this study, the data was collected after obtaining consent from the respondents, and the data was collected from normal people, not patients.

Reference

  1. Hair, J.F.; Jr., Black, W.C.; Babin, B.J.; Anderson, R.E.; Tatham, R.L. Multivariate data analysis (6th ed.). Upper Saddle River, NJ: Prentice-Hall, 2006.

  1. I suggest to rewritten the paper and to analyze selected problems, not just to immerse in the statistical issues, but remember about the main question.

Response: Thank you for your all comments. We revised our paper according to your valuable comments.

Again, we truly appreciate your time and help. Your comments and suggestions definitely have improved the quality of this manuscript.

Reviewer 3 Report

This paper investigated experience economy, brand attitudes, and brand loyalty in the restaurant industry both in robot-serving and human-serving restaurants. I recommend the authors make small changes as suggested below.

In the 4.3. Structural equation modeling on page 8, the authors said that all hypotheses were supported, however they should thoroughly discuss the findings of the study. As presented in Figure 2, the effect of esthetics was not statistically supported in RR model, the influence of education was not supported in RH model, and the impact of education was not supported in MTD model. Please discuss these findings thoroughly in this section.

Concerning 5.2. Managerial implications, I would recommend that the authors discuss how restaurant mangers can manipulate entertainment and escapism elements for each restaurant type (robot-serving and human-serving restaurants). Specific practical implications and examples for each type should be provided for restaurant managers or marketers. 

In addition, please discuss how managers can apply the findings related to the effect of education in restaurants where robot provides services. How can restaurants can arouse customers' curiosity and propine learning experience? Please present specific examples. Also discuss how managers can design differentiated service experience utilizing esthetics element in human serving restaurants. 

In terms of practical implications, the authors should make  more efforts to provide specific examples that the restaurant managers could apply in their establishment, which could be the main contribution of the study.

Author Response

Reviewer 3

  1. This paper investigated experience economy, brand attitudes, and brand loyalty in the restaurant industry both in robot-serving and human-serving restaurants. I recommend the authors make small changes as suggested below.

Response: We sincerely appreciate your important comments and suggestions on the previous version of this manuscript. Our responses to your comments are summarized in the following section.

  1. In the 4.3. Structural equation modeling on page 8, the authors said that all hypotheses were supported, however they should thoroughly discuss the findings of the study. As presented in Figure 2, the effect of esthetics was not statistically supported in RR model, the influence of education was not supported in RH model, and the impact of education was not supported in MTD model. Please discuss these findings thoroughly in this section.

Response: Thank you for this critical comment. As you suggested, we clearly explained the results of data analysis as follows:

☞ The data analysis indicated that education positively affects brand attitude in RR, however there is no relationship between the two concepts in RH and MTD. Therefore, Hypothesis 1 was partially accepted. Second, entertainment has a positive influence on brand attitude in RR, RH, and MTD, so Hypothesis 2 was accepted. Third, the effect of esthetics on brand attitude was identified, except for RR. Hence, Hypothesis 3 was partially accepted. Fourth, escapism plays an important role in the formation of brand attitude in the all three models, so Hypothesis 4 was accepted. Lastly, the relationship between brand attitude and brand loyalty was found in RR, RH, and MTD, Thus, Hypothesis 5 was accepted.

  1. Concerning 5.2. Managerial implications, I would recommend that the authors discuss how restaurant mangers can manipulate entertainment and escapism elements for each restaurant type (robot-serving and human-serving restaurants). Specific practical implications and examples for each type should be provided for restaurant managers or marketers.

Response: Thank you for your critical comment. As you suggested, we added specific practical implications and examples for each type, focusing on entertainment and escapism elements as follows:

☞ Entertainment is the most important factor in forming brand attitude in RR. As described earlier, entertainment can be defined as the act of amusing or entertaining people. Zhang, Gursoy, and Shi [53] suggested robots should be augmented to have the ability to make humorous responses and use witty language in order to improve entertainment services. That is, robots should be able to engage in humorous language when interacting with consumers rather than just a typical service language. For instance, when consumers order spicy food, the robot can say, “Welcome to the spicy food challenge! Hey buddy, you should be ready to call 911” instead of just taking orders. In addition, escapism also aids to enhance brand attitude in RR. Marketers at RR should provide escapism experiences as consumers have a lot of stress due to COVID-19. Consumers prefer non-face-to-face services to prevent infection after the outbreak of the pandemic [9]. For instance, marketers can entice consumers with messages such as “Enjoy your gourmet life without worrying about the pandemic” and provide completely non-face-to-face services by robots. By recognizing these advantages, consumers would have a comfortable experience as if they have escaped from social distancing life when they visit RR.

Also, in the RH, entertainment was the most significant factor in affecting brand attitude. Thus, RH requires making entertainment service strategies differently when compared with RR. For example, Etude House, which is a cosmetic brand, welcomes consumers with its interesting concept and princess aesthetic [55]. More specifically, the service concept is that the employees are the princesses of Etude Kingdom and consumers are neighboring countries’ princesses, thus they call consumers “Princess”. In other words, the unique service concept provides entertainment services with a fun conversation manual tailored to the brand. For instance, managers can compose a service manual with a dialogue that appropriately mixes either dialect or language of the country when serving cuisine from the specific country.

  1. In addition, please discuss how managers can apply the findings related to the effect of education in restaurants where robot provides services. How can restaurants can arouse customers' curiosity and propine learning experience? Please present specific examples. Also discuss how managers can design differentiated service experience utilizing esthetics element in human serving restaurants.

Response: Thank you for your critical comment. As you suggested, the following paragraphs were added in order to show important managerial implications of this study related to education and esthetics

☞ Lastly, education is also a significant variable in forming brand attitudes in RR. Therefore, robot servers should provide more educational experiences. Sauppé and Mutlu [54] said the robot’s screen, which doubles as a face, can offer more information. For instance, the robot servers can efficiently inform consumers about culinary processes and menu descriptions on their screen when serving menus. These robot service experience strategies will aid to enhance RR's brand attitude and improve brand loyalty.

In RR, education, entertainment, and escapism aid to increase brand attitude. In addition, in RH, entertainment, esthetics, and escapism help to enhance brand attitude. Thus, the results suggest practical strategies for each restaurant type as follows.

In addition, human servers’ esthetics had a positive effect on brand attitude toward RH, and it was not supported by RR. Despite the issue of the burden of esthetic on employees [56], this is still an important element in hospitality industries. Managers should consider refined uniforms and create a service manual for elegant gestures and service behaviors, excluding personal esthetic elements such as personal grooming, hairstyle, and hair length. Lastly, escapism is also a significant variable in forming brand attitudes in RH. Consumers want to have new experiences which make them feel they have escaped from their daily routines [57]. Managers should design serving processes for a special experience that can only be provided by professional human servers, such as flambé and bartending performances on consumers' tables. These experience strategies would improve brand attitude toward RH and aid to enhance consumers’ loyalty.

  1. In terms of practical implications, the authors should make more efforts to provide specific examples that the restaurant managers could apply in their establishment, which could be the main contribution of the study.

Response: Thank you for your comment. As you suggested, we rewrote managerial implications in order to clearly explain practical strategies. Please refer to our response for your earlier comment.

This was a very thorough and impressive review. We feel that our manuscript has been much improved because of your contribution. We truly appreciate it.

Round 2

Reviewer 2 Report

Dear Authors,

thank you for your responses and modifications of the paper. Now, after changing the title, the content fits to the topic and it is more clear for the reader. If you have such ethical regulations and the journal publisher accepts them, then I have no objection.  When writing about p-fishing, I used the metaphor that many researchers make the mistake of looking for correlations instead of focusing on the chosen problem and describing it accurately. Multidimensional influences on something are rather the domain of trend research. One selected research problem, well described and explained brings more new to the understanding of the problem than describing vague multiple correlations. Each such thread, by itself, could be the subject of a separate article, precisely because of the complexity of the issue. An article is rather a succinct discussion of a problem, as opposed to a book, which provides adequate opportunities to develop a complex problem. The changes made, however, have made me feel positive about the effect of your work and I appreciate your contribution to science

Kind regards,

The Reviewer